# OpenReview forum: "Tree-based Dialogue Reinforced Policy Optimization for Red-Teaming Attacks"
_ICLR.cc/2026/Conference — ICLR 2026 Poster_

### Official Review · Reviewer_HvSg · 2025-10-25

**Soundness:** 3
**Presentation:** 2
**Contribution:** 2
**Rating:** 4
**Confidence:** 4

**Summary:**

This paper introduces DIALTREE-RPO, a novel on-policy reinforcement learning framework that automates the discovery of multi-turn jailbreak strategies against large language models. It treats red-teaming as a goal-oriented, sequential decision-making problem, using tree-structured dialogue exploration and policy optimization to learn adaptive attack strategies without human-crafted templates or data.

**Strengths:**

1.	First to frame multi-turn red-teaming as a sequential decision problem; introduces a tree-search + GRPO framework that eliminates the need for human-written templates.
2.	Achieves 85% average ASR across 10 models—26% better than the prior best—with only 3 queries per attack; experiments are thorough and reproducible.
3.	Exposes a systemic LLM vulnerability to strategic multi-turn attacks; releasing the methodology (not payloads) accelerates the development of multi-turn safety defenses.

**Weaknesses:**

1. Lack of novelty. Tree-search + RL for dialogue is not virgin territory. TAP already combines tree expansion + pruning for jailbreaks;
2. This paper did not propose any new jailbreak datasets either; instead, it reused the datasets provided by MTSA.
3. No tests have been conducted to evaluate the robustness of this jailbreaking method under different defenses.

**Questions:**

1. There is no report on the performance of the red team model after the cold start, nor is there any information on how much the RL has improved it.
2. No tests were conducted to evaluate the robustness of this jailbreaking method under different defenses, which will affect the practical application of the jailbreaking method.
3. The paper claims that "it autonomously discovers diverse multi-round attack strategies". Are there any data to support this statement?

**Details Of Ethics Concerns:**

The paper proposes an automated multi-round attack method, which significantly improves the success rate of attacks on mainstream models. However, it lacks a responsible release mechanism (such as weight control and coordinated disclosure), and plans to publicly release the training data and code containing detailed harmful content, which poses a risk of being directly weaponized.

---

> ### Author Response · Authors · 2025-11-26
> **Official Responses to Reviewer HvSg [1/2]**
>
> We are grateful for the reviewer's constructive comments. We respond to each of your comments in the following.
>
> **W1: Lack of novelty.**
>
> We would like to clarify that our work’s technical contributions are beyond the simplistic combination of tree-search + RL with dialogue, and our approach substantially differs from TAP. **First,** most existing tree-search RL frameworks are restricted to the reasoning tasks that are non-interactive and have verifiable rewards. Our framework is the first to propose an **on-policy RL** framework with dialogue tree rollout for multi-turn red-teaming with non-verifiable and proxy-based rewards. **Second,** TAP is an inference-time search method that utilizes a tree to find a prompt without any learning process, whereas our DialTree-RPO uses tree-structured exploration to learn an attacker policy that can discover more diverse and effective jailbreaking strategies. **Third,** beyond proposing Dialogue Tree Rollout with Pruning, we further propose a novel adaptive masking mechanism to resolve the format unlearning issue in multi-turn RL.
>
> **W2: Did not propose new jailbreak datasets.**
>
> We would like to clarify that our primary contribution is a novel RL framework for multi-turn red-teaming integrated with tree-based rollout and adaptive masking, rather than curation of datasets. In fact, one of our contributions is our **superior data efficiency**. For instance, we only utilize 500 training instances (MTSA uses 2073 instances) and achieve **28.1%** ASR better performance. This demonstrates that DialTree-RPO can automate the discovery of diverse attack strategies without requiring labor-intensive curation of new and large-scale datasets.
>
> **W3: No tests to evaluate the robustness under different defenses.**
>
> We clarify that the targeted LLMs in our experiments, including both open-source and closed-source models, are mostly post-trained with safety alignment, which represents the primary defense deployed in practice. Closed-source models such as GPT-4o may additionally employ inference-time guardrails and content moderation since they are accessed through API calls.
>
> To further evaluate the robustness of our method, we conduct an additional experiment to test whether our approach can jailbreak the systems equipped with specialized safety guardrails. Specifically, based on the original target model, we additionally employ prompt classification on the attacker’s input and response classification on the target models’ outputs. The system is considered jailbroken if and only if both prompt and response classification from the guardrail predict “safe” (i.e., the guardrail is bypassed), AND the target model indeed elicits harmful responses (determined by the GPT-4o judge).
>
> Specifically, we compare our method against the SFT baseline across three target models paired with their corresponding guardrails: Llama-3.1-8B with Llama-Guard-3-8B, Gemma-2-9B with ShieldGemma-9B, and GPT-4.1-mini with GPT-oss-safeguard-20B. As shown in Table A, our method consistently outperforms the SFT baseline across all configurations. Notably, while the SFT baseline experiences a substantial performance drop when guardrails are introduced, our method maintains relatively high attack success rates. These results demonstrate that our approach is robust against additional safety guardrails and can effectively bypass both the target model's inherent safety alignment and external defense mechanisms.
>
> **Table A: Attack success rate (%) with and without external guardrails.**
>
> | **Method** | **Llama-3.1-8B (w/o Guardrail)** | **Llama-3.1-8B (w/ Llama-Guard-3-8B)** | **Gemma-2-9B (w/o Guardrail)** | **Gemma-2-9B (w/ ShieldGemma-9B)** | **GPT-4.1-mini (w/o Guardrail)** | **GPT-4.1-mini (w/ GPT-oss-safeguard-20B)** |
> |--------|:----------------------------:|:----------------------------------:|:--------------------------:|:------------------------------:|:----------------------------:|:---------------------------------------:|
> | SFT | 51.0 | 20.0 | 50.5 | 37.5 | 53.5 | 45.0 |
> | **DialTree-RPO (Ours)** | 81.5 | 76.5 | 83.0 | 61.5 | 90.0 | 85.5 |

---

> ### Author Response · Authors · 2025-11-26
> **Official Responses to Reviewer HvSg [2/2]**
>
> **Q1: Lack of ablation on the SFT stage.**
>
> We thank the reviewer for this important question. We have conducted the requested ablation comparing two models: (1) SFT-only baseline and (2) our full DialTree-RPO.
>
> **Table B: The comparison between the SFT-only baseline and DialTree-RPO in terms of ASR (%).**
>
> | | GPT-4o | GPT-4.1-mini | o3-mini | Gemini-2.0-Flash | Grok-4 | Llama-3.1-8B | Llama-3.3-70B | Mistral-7B | Gemma-2-2B | Gemma-2-9B | Avg. |
> |:---:|:------:|:------------:|:-------:|:----------------:|:------:|:---------------:|:----------------:|:------------------------:|:-------------:|:-------------:|:-------:|
> | **SFT-Only** | 44.5 | 53.5 | 22 | 48.5 | 7.5 | 51 | 60.5 | 58 | 43 | 50.5 | 43.9 |
> | **DialTree-RPO (Ours)** | 86 | 90 | 86.5 | 87.5 | 75 | 81.5 | 89.5 | 85 | 88.5 | 83 | 85.25 |
> | **Δ** | 41.5 | 36.5 | 64.5 | 39 | 67.5 | 30.5 | 29 | 27 | 45.5 | 32.5 | 41.35 |
>
> From Table B, our proposed RL stage consistently contributes significant improvements over the SFT-only baseline across all 10 target models, with an average gain of 41.35%. This shows that while SFT provides the necessary initialization (e.g., learning format-following and basic jailbreaking capabilities), our proposed RL stage is crucial for discovering more effective attack strategies. We have included the experiment in Appendix D.7 and Table 9 in our revised paper.
>
> **Q2**: See the response in the W3 above.
>
> **Q3: Evidence of autonomous discovery of new attack strategies**
>
> We provided evidence for this claim in Section 5 and Appendix G.1. Through manual analysis of **60** randomly sampled dialogues, we identified **more than 6** new attack strategies discovered by our approach **that are absent from the SFT training data**, such as cross-lingual evasion (interleaving multiple languages to obscure harmful intent and evade detection), jigsaw attack (iteratively eliciting isolated components that assemble into complete harmful information), incremental refinement (i.e., requesting iterative improvements to a borderline response until it crosses safety thresholds), sycophancy exploitation (i.e., expressing disappointment to pressure the model into compliance), and so on. These findings demonstrate that DialTree-RPO can autonomously discover novel red-teaming strategies beyond its training distribution. We believe this analysis also provides actionable insights for developing targeted countermeasures.
>
> **Q4 [Response to ethical concern]: Lacks a responsible release mechanism**
>
> We thank the reviewer for raising this important question. To prevent the malicious misuse of our DialTree-RPO, we plan to conduct rigorous and responsible disclosure practices. We will provide access to trained attacker models, code, and data upon request to verified researchers who demonstrate legitimate safety research objectives and agree to use the models exclusively for defensive applications without redistribution. Access requests can be submitted by emailing the authors with their institutional affiliation and research proposal, and they will be meticulously assessed by the authors. We will carefully revise the Ethics Statement section to address your ethics concerns.

---

> > ### Comment · Reviewer_HvSg · 2025-11-26
> > **Thank the authors for the response**
> >
> > Thank the authors for the response in the rebuttal.  I have carefully read the authors' rebuttal and plan to maintain my original score.

---

> > > ### Author Response · Authors · 2025-11-26
> > >
> > > Dear Reviewer HvSg,
> > >
> > > Thank you for taking the time to carefully read our rebuttal. We truly appreciate your engagement with our work.
> > >
> > > To help us better understand how we might strengthen the paper, could you share which specific concerns remained after our response? We are genuinely committed to improving our work and would value any concrete feedback from you.
> > >
> > > Thank you again for your time and constructive feedback.

---

### Official Review · Reviewer_hWje · 2025-10-30

**Soundness:** 3
**Presentation:** 3
**Contribution:** 3
**Rating:** 6
**Confidence:** 4

**Summary:**

This paper introduces DIALTREE-RPO, a reinforcement learning framework that formulates multi-turn red-teaming as sequential decision-making. The method uses tree-based exploration with quality-aware pruning to efficiently search attack trajectories, and adaptive masking to prevent format degradation during training. Trained against a single small model (Llama-3.2-1B) using on-policy GRPO, it achieves 85.3% average attack success rate across 10 diverse target models including GPT-4o and Gemini-2.0, outperforming prior methods by 25.9% with fewer queries. The framework autonomously discovers novel attack strategies like gradual escalation and cross-lingual evasion, demonstrating that multi-turn interactions expose significantly greater vulnerabilities than single-turn attacks in current LLMs.

**Strengths:**

Formulates multi-turn red-teaming as a goal-oriented sequential decision-making problem, extending GRPO to dialogues with non-verifiable rewards.

Dialogue tree rollout with quality-aware pruning enables structured exploration of exponentially large attack spaces, while adaptive masking solves the critical format unlearning problem.

Tests against diverse closed-source (GPT-4o, Gemini-2.0, o3-mini) and open-source models; includes thorough ablations on tree depth, branching factor, group size, and pruning strategies; validates automated evaluation with human agreement.

**Weaknesses:**

Although the paper cites previous state-of-the-art methods like X-Teaming (https://arxiv.org/abs/2504.13203
), which shows a higher ASR (94.3%) compared to this paper (86%), it lacks a comparison with prior multi-turn state-of-the-art methods such as X-Teaming and ActorAttack (https://arxiv.org/abs/2410.10700).

The paper primarily focuses on the attack side and does not explore the defense side.

**Questions:**

Since previous papers like X-Teaming and ActorAttack show higher effectiveness, does DIALTREE-RPO offer any other advantages, such as improved efficiency or diversity?

Is there analysis done on Claude 3.7/4 Sonnet model, which has been considered nearly immune to
single-turn attacks (https://arxiv.org/pdf/2501.18837)

Table 4 shows HarmAug-Guard has 84.73% accuracy with 27% false positive rate on JailbreakBench. How do reward model errors affect policy learning?

What responsible disclosure and access control practices will be implemented to prevent the malicious use of DIALTREE-RPO?

---

> ### Author Response · Authors · 2025-11-26
> **Official Responses to Reviewer hWje [1/2]**
>
> We appreciate the reviewer's detailed and constructive feedback. We respond to each question as follows.
>
> **W1: Lacking a comparison with X-Teaming and ActorAttack.**
>
> We thank the reviewer for this suggestion. We have conducted additional experiments comparing with ActorAttack and X-Teaming following their official implementations. Note that for fair comparison, we adopt a consistent evaluation setup across our approach and baselines, where each example is evaluated with a single jailbreaking attempt (denoted as ASR@1). We report the ASR@1 on 4 target models (i.e., GPT-4o, GPT-4.1-mini, Llama-3.1-8B, and Gemma-2-9B) in Table A, and the average ASR on 4 target models with 1, 3, and 5 attempts in Table B.
>
> **Table A: Performance comparison with ActorAttack and X-Teaming in terms of ASR (%) on HarmBench.**
>
> | **Method** | **GPT-4o** | **GPT-4.1-mini** | **Llama-3.1-8B** | **Gemma-2-9B** | **Avg. ASR@1** |
> |---|:---:|:---:|:---:|:---:|:---:|
> | ActorAttack | 25.5 | 35 | 12 | 18 | 22.6 |
> | X-Teaming | 48 | 54.5 | 43 | 34 | 44.9 |
> | DialTree-RPO (Ours) | **86** | **90** | **81.5** | **83** | **85.1** |
>
> **Table B: Test-time scaling performance: Average ASR (%) with N attempts (ASR@N) across 4 target LLMs, i.e., GPT-4o, GPT-4.1-mini, Llama-3.1-8B, and Gemma-2-9B, on HarmBench.**
>
> | **Method** | **ASR@1** | **ASR@3** | **ASR@5** |
> |---|:---:|:---:|:---:|
> | ActorAttack | 22.6 | 38.5 | 45.1 |
> | X-Teaming | 44.9 | 69.6 | 78.9 |
> | DialTree-RPO (Ours) | **85.1** | **98.6** | **99.5** |
>
> As shown in Tables A and B, our method significantly outperforms both baselines across all target models. We note that ActorAttack’s originally reported ASR is based on ASR@3 and X-Teaming's is based on ASR@10, whereas our evaluation uses ASR@1 for all baselines and DialTree-RPO for fair comparison. With test-time scaling, our method reaches near-perfect success rates (**99.5%**) at just 5 attempts, surpassing X-Teaming's ASR@10 performance. Furthermore, X-Teaming incurs substantial computational costs due to its multi-agent framework. In our experiments, evaluating X-Teaming at ASR@5 costs approximately $79.8 for each target model, whereas our method solely uses an 8B attacker model and does not incur any API cost. These results demonstrate that DialTree-RPO offers substantial advantages in both effectiveness and efficiency. We have included the results and analysis in Appendix D.11.
>
> **W2: Does not explore the defense side.**
>
> We appreciate this comment. While defense mechanisms are important for AI safety, our paper focuses on establishing a new RL framework for multi-turn red-teaming to systematically uncover safety vulnerabilities in LLMs. The defense-side research constitutes a separate direction requiring different methodologies and would substantially go beyond our current paper’s scope. We leave exploring more advanced defense approaches for future work.
>
> It is worth noting that we provide substantial actionable insights for defense research, where we conducted a comprehensive analysis of multi-turn attack strategies uncovered by DialTree-RPO in Appendix G.1. This thorough analysis can enlighten future research to develop targeted countermeasures against these attack patterns. We have extended the discussion of potential defenses in Appendix F.

---

> ### Author Response · Authors · 2025-11-26
> **Official Responses to Reviewer hWje [2/2]**
>
> **Q1**: See the response in W1 above.
>
> **Q2: Additional Experiments on Attacking Claude.**
>
> We thank the reviewer for raising this important point. We have conducted additional experiments with Claude-4-Sonnet as the target model. As shown in Table C, our DialTree-RPO achieves 71% ASR@1 and 96.5% ASR@5, substantially outperforming the SFT baseline and X-Teaming. These results demonstrate that our approach can effectively jailbreak even one of the most safety-aligned models, where existing methods largely fail. We have included these results in the revision.
>
> **Table C: Red-teaming results on Claude-4-Sonnet in terms of ASR (%).**
>
> | **Metric** | **SFT Baseline** | **X-Teaming** | **DialTree-RPO (Ours)** |
> |-----------------|:-----:|:-----------:|:----------:|
> | ASR@1 | 1.5 | 9.5 | **71** |
> | ASR@5 | 4 | 32.5 | **96.5** |
>
> **Q3: How do reward model errors affect policy learning**
>
> We thank the reviewer for raising this question. Based on our empirical observations, weaker reward models typically lead to unstable training, slower convergence, and larger variance, as RL becomes weakly-supervised under noisy reward signals. However, our experimental results show that the errors in our reward model are minor, and our DialTree-RPO is robust to this noise. Figure 2(c) shows a steady and stable reward increase throughout training, indicating that our policy converges effectively without high variance. Furthermore, we validate actual attack success using a held-out GPT-4o judge and human evaluation on the test set (Appendix D.2), ensuring that our reported performance reflects true effectiveness rather than reward model artifacts.
>
> **Q4: What will be implemented to prevent malicious use**
>
> We thank the reviewer for raising this important question. To prevent the malicious misuse of our DialTree-RPO, we plan to conduct rigorous and responsible disclosure practices. We will provide access to trained attacker models, code, and data upon request to verified researchers who demonstrate legitimate safety research objectives and agree to use the models exclusively for defensive applications without redistribution. Access requests can be submitted by emailing the authors with their institutional affiliation and research proposal, and they will be meticulously assessed by the authors.

---

### Official Review · Reviewer_63Yg · 2025-11-01

**Soundness:** 3
**Presentation:** 3
**Contribution:** 2
**Rating:** 4
**Confidence:** 4

**Summary:**

The paper shows an RL post-training method for multi-turn red-teaming attacks via sequential decision-making in the dialogue. The RL training pipeline leverates exploration instead of labeled data collection. Experiments show the higher ASR of the proposed method compared to baselines.

**Strengths:**

The paper is generally well-organized, and the figures are clear to follow. The problem of RL post-training in a multi-turn setting is sound and important. It is also interesting to regard the multi-turn setting as a form of decision-making in dialogue in which RL can be applied.

**Weaknesses:**

- Overall, the method is based on GRPO by applying it to the multi-turn dialogue setting, which is a little bit incremental regarding novelty and contribution. One possible way is to conduct an experiment based on other RL post-training methods, e.g., RLOO, REINFORCE++, DAPO.

- In the node pruning process, since the previous node will be pruned according to the same criteria of topic adherence, the context will be the same along all sampled trajectories along the valid tree nodes and the diversity of the trajectories will be influenced a lot. One possible way to mitigate it might be to develop an adaptive branching factor to make it more diverse. Also, the branching factor needs to be larger in Figure 3, e.g., 16, 32.

- In the reward design, it uses a classifier to construct a verifiable reward for this non-verifiable problem, as stated in the beginning. However, it is still at risk of reward hacking if the judge is not perfect.

- The adaptive masking part needs more formal presentation, which is expected to be shown in Eq. (3), for example. Figure 2 uses running smoothing for presentation, which is different from the standard deviation or confidence interval. More experiments with different seeds are expected.

- More related work on multi-turn jailbreak defense (e.g. [1] [2]) needs to be discussed in Sec 6 for comprehensiveness, especially [2] seems to also explicitly model the multi-turn dialogue as sequential decision-making in the context of jailbreaks.

---
[1] Lu et al. X-Boundary: Establishing Exact Safety Boundary to Shield LLMs from Multi-Turn Jailbreaks without Compromising Usability, arxiv, 2025

[2] Hu et al. Steering Dialogue Dynamics for Robustness against Multi-turn Jailbreaking Attacks, arxiv, 2025

**Questions:**

See Weakness

---

> ### Author Response · Authors · 2025-11-26
> **Official Responses to Reviewer 63Yg [1/2]**
>
> We thank the reviewer for their valuable feedback and thoughtful suggestions. We address each comment as follows.
>
> **W1: A little bit incremental regarding novelty and contribution.**
>
> We would like to justify that our work is beyond simply applying GRPO to multi-turn dialogue. Our core contributions consist of three key aspects:
> **(1)** To our best knowledge, we are the first to propose on-policy RL specifically for multi-turn jailbreaking, whereas existing works mostly rely on off-policy methods or do not incorporate a learning process.
> **(2)** Unlike standard GRPO which samples independent linear trajectories, our DialTree-RPO enables structured exploration of attack strategies, along with quality-aware pruning based on format validity and topic adherence;
> **(3)** Through our empirical analysis, we identify a critical challenge in multi-turn conversational RL, i.e., format unlearning. We introduce a novel adaptive masking technique to selectively protect format tokens in negative-advantage trajectories and significantly improve training stability.
>
> In addition, we apply our proposed method to another RL algorithm, i.e., DAPO [1]. We adopt the same training configuration as the one we used for GRPO. Due to computation and time limitations, we used 200 goals for training and set the branching factor to 2. From Table A, our DialTree-RPO with DAPO also achieves substantial improvement compared with the SFT baseline. This result demonstrates that our approach can generalize to other RL methods with similar performance gain.
>
> **Table A: Results of DialTree-RPO with GRPO and DAPO in ASR (%).**
>
> |**Method** | **Llama-3.1-8B** | **Mistral-7B** | **Gemma-2-9B** | **Avg.** |
> |----------------------|:--------------:|:------------:|:------------:|:-------:|
> | SFT Baseline | 51 | 58 | 50.5 | 53.2 |
> | DialTree-RPO (GRPO) | **71.5** | 86.5 | **70.5** | **76.2** |
> | DialTree-RPO (DAPO) | 67.5 | **88** | 66 | 73.8 |
>
> **W2:  Diversity of the trajectories will be influenced.**
>
> As we described in lines 170-171 and lines 862-863, the off-topic pruning aims to mitigate significant conversational drift from the goal. The pruned off-topic branches will not affect the diversity of trajectories, since these branches are low-quality and will introduce noise in the training (demonstrated by the ablation study in Table 2).
>
> We also include an additional analysis to investigate how the branching factor n affects diversity. We follow prior work (Guo et al., 2025c) and use self-BLEU [2] to monitor the diversity of trajectories during training.
>
> **Table B: Impact of the branching factor n on trajectory diversity. We report the average self-BLEU scores across the first 100 RL training steps. The lower self-BLEU indicates the higher diversity.**
>
> | **Branching Factor** | **n=1** | **n=2** | **n=4** | **n=8** |
> |-----------|:-----:|:-----:|:-----:|:-----:|
> | **Self-BLEU (&darr;)** | 0.554 | 0.221 | **0.160** | 0.269 |
>
> From Table B, our method can already achieve sufficient diversity when n=4. When n is increased to 8, the diversity drops instead as semantically similar examples emerge. In addition, a larger branching factor will increase the computational cost exponentially, so we did not explore the settings with n>= 16. We acknowledge that adaptive branching is an interesting direction and will note this as future work. We have included the results in Appendix D.9.
>
> **W3: Risk of reward hacking.**
>
> We acknowledge that reward hacking is indeed an important consideration in any RL frameworks with non-verifiable rewards, and this remains an active research challenge in the community. However, we have taken several steps to mitigate this risk:
>
> **(1)** We carefully selected HarmAug-Guard as our reward model through extensive benchmarking (shown in Appendix D.1), showing it achieves a strong balance between performance and efficiency.
>
> **(2)** We evaluate our trained attacker using a different LLM judge (i.e., GPT-4o) at test time, which demonstrates our method is genuinely effective rather than exploiting specific artifacts in the reward model.
>
> **(3)** Our human evaluation shown in Appendix D.2 shows substantial agreement (w/ Cohen's Kappa = **74.7%**) between GPT-4o and human annotators, further validating that our automated evaluation reflects real harmfulness. While we cannot eliminate reward hacking entirely, these measures provide strong evidence that our method learns robust attack strategies. We have included the discussion on the risk of reward hacking in Appendix F.

---

> ### Author Response · Authors · 2025-11-26
> **Official Responses to Reviewer 63Yg [2/2]**
>
> **W4: Better presentation for adaptive masking and robustness analysis of DialTree-RPO.**
>
> We have updated Eq. (3) and (4) to provide a formal definition of the adaptive masking mechanism. In addition, we conducted 5 independent training runs using different random seeds under the same training configuration as the main experiments. Table C below reports the mean and standard deviation of ASR across five runs. The results confirm that our method consistently outperforms the baseline across all runs, demonstrating both reproducibility and robustness to different seeds. We have included this table in Appendix D.10 and Table 12.
>
> **Table C: Mean and the standard deviation of ASR (%) across 5 training runs.**
>
> | **GPT-4o** | **GPT-4.1-mini** | **o3-mini** | **Gemini-2.0-Flash** | **Grok-4** | **Llama-3.1-8B** | **Llama-3.3-70B** | **Mistral-7B** | **Gemma-2-2B** | **Gemma-2-9B** |
> |:----------:|:---------------:|:-----------:|:-------------------:|:----------:|:---------------:|:----------------:|:--------------:|:--------------:|:--------------:|
> | 85.8 ± 6.13 | 93.80 ± 2.97 | 87.83 ± 8.02 | 89.88 ± 3.35 | 90.37 ± 10.82 | 81.40 ± 5.32 | 94.70 ± 5.74 | 91.80 ± 6.70 | 89.20 ± 5.25 | 81.20 ± 6.88 |
>
> **W5: Comparison with more relevant works.**
>
> We appreciate the pointer to these recent works on multi-turn jailbreak defense. We added citations to Lu et al. and Hu et al. in Section 6 and clarified how our work differs in our updated paper. Both papers focus on defense mechanisms against multi-turn attacks, while we develop an attack method to discover vulnerabilities. Particularly, while Hu et al. also model the multi-turn jailbreaking dynamics, they primarily utilize it for developing a defense strategy, and the modeling does not involve sophisticated learning processes. On the contrary, we focus on building a multi-turn RL framework to model the complex sequential decision-making and discover safety vulnerabilities of target LLMs via jailbreaking.
>
> ---
>
> **References**
>
> [1] DAPO: An Open-Source LLM Reinforcement Learning System at Scale. NeurIPS 2025.
>
> [2] Texygen: A Benchmarking Platform for Text Generation Models. SIGIR 2018.

---

> > ### Comment · Reviewer_63Yg · 2025-11-26
> >
> > Thank you for the detailed rebuttal and clarification. My concerns have been addressed and the additional experiment results should be reflected in the updated version. I will raise my score to 6. Good luck!

---

> > > ### Author Response · Authors · 2025-11-26
> > >
> > > Thank you very much for raising the score and for your thoughtful engagement! We sincerely appreciate the time you invested in the reviewing process and your recognition of our work.

---

### Official Review · Reviewer_uVB4 · 2025-11-04

**Soundness:** 3
**Presentation:** 2
**Contribution:** 2
**Rating:** 2
**Confidence:** 4

**Summary:**

This paper introduces DIALTREE-RPO, a tree-based reinforcement learning framework for multi-turn red-teaming attacks against LLMs. By modeling adversarial dialogues as sequential decision making and using structured exploration with pruning and adaptive masking, the method autonomously discovers diverse multi-turn attack strategies that outperform previous baselines in attack success rate and efficiency.

**Strengths:**

1. The paper addresses an important gap in AI safety: the vulnerability of LLMs to adaptive multi-turn attacks.
2. The tree-based exploration and pruning mechanisms are conceptually well-motivated for discovering diverse trajectories, which is crucial for efficient red-teaming.
3. The empirical results are strong and comprehensive, showing clear improvements over previous baselines across a variety of models.
4. The proposed method has promising transferability, being able to jailbreak much larger models after training on smaller ones.

**Weaknesses:**

1. **Lack of ablation on SFT stage:** There is no explicit comparison of attack efficacy using only the SFT (supervised fine-tuning) attacker before RL training. The improvement brought by RL over the SFT baseline is unclear, which is an important ablation for understanding the contribution of each stage.
2. **Pruning criteria are overly simplistic:** The current pruning process only eliminates obviously problematic branches, such as those with malformed outputs or irrelevant topics. However, it does not attempt to distinguish among the valid branches to identify which ones have higher potential for successful attack. As a result, promising attack trajectories may be overlooked, since the approach only performs random subsampling among the remaining candidates rather than scoring or prioritizing those with greater promise. A more sophisticated strategy—such as utilizing PRMs to rank and select the most promising branches—could help ensure that the tree search focuses resources on the most effective attack paths.
3. **Justification for adaptive masking is unclear:** The effectiveness of adaptive masking is attributed to format retention. However, since the reward is not directly related to format, the sign of the advantage may not reflect format quality, so I'm very curious why this happens. The experimental section also lacks a thorough analysis explaining why masking format tokens on negative-advantage trajectories is beneficial.
4. **Reward and evaluation inconsistency:** The training stage uses HarmAugGuard for harmfulness evaluation, while the test stage resorts to GPT-4o as the judge—these may be misaligned.
5. **Ambiguity in tree rollout vs. GRPO comparison:** In Table 2 and line 370, the comparison between "w/o any pruning" (tree rollout only) and "w/o tree rollout" (standard GRPO) is unclear and seemingly contradictory—the results show that GRPO performs better than tree rollout-only, yet the text claims better gain from tree rollout than GRPO. The statement requires clarification.
6. **Utterance length limitation:** In Appendix C.2, you mention limiting outputs to 256 and 150 tokens. How is this enforced? Is it simple truncation? RL training can lead to longer outputs; could this be responsible for the increase in malformed outputs in Figure 2, i.e., the model may not have finished its format before truncation?

**Questions:**

1. **Tree rollout path:** With a maximum depth of 5 and a branching factor of 2, the maximum possible number of paths is 32. Is supplemental sampling performed when the actual number of paths falls short due to pruning?
2. **Jailbreak-R1 implementation:** Jailbreak-R1 is a single-turn method. How did you use it in your multi-turn setup? Was it modified for multi-turn, or was a single-turn evaluation used?

---

> ### Author Response · Authors · 2025-11-26
> **Official Responses to Reviewer uVB4 [1/2]**
>
> We appreciate the reviewer for their time and effort to provide insightful comments. Our responses to your concerns are as follows.
>
> **W1: Lack of ablation on the SFT stage.**
>
> We thank the reviewer for this important question. We have conducted the requested ablation comparing two models: (1) SFT-only baseline and (2) our full DialTree-RPO.
>
> **Table A: The comparison between the SFT-only baseline and DialTree-RPO in terms of ASR (%).**
>
> | | GPT-4o | GPT-4.1-mini | o3-mini | Gemini-2.0-Flash | Grok-4 | Llama-3.1-8B | Llama-3.3-70B | Mistral-7B | Gemma-2-2B | Gemma-2-9B | Avg. |
> |:---:|:------:|:------------:|:-------:|:----------------:|:------:|:---------------:|:----------------:|:------------------------:|:-------------:|:-------------:|:-------:|
> | **SFT-Only** | 44.5 | 53.5 | 22 | 48.5 | 7.5 | 51 | 60.5 | 58 | 43 | 50.5 | 43.9 |
> | **DialTree-RPO (Ours)** | 86 | 90 | 86.5 | 87.5 | 75 | 81.5 | 89.5 | 85 | 88.5 | 83 | 85.25 |
> | **Δ** | 41.5 | 36.5 | 64.5 | 39 | 67.5 | 30.5 | 29 | 27 | 45.5 | 32.5 | 41.35 |
>
> From Table A, our proposed RL stage consistently contributes significant improvements over the SFT-only baseline across all 10 target models, with an average gain of 41.35%. This shows that while SFT provides the necessary initialization (e.g., learning format-following and basic jailbreaking capabilities), our proposed RL stage is crucial for discovering more effective attack strategies. We have included the experiment in Appendix D.7 and Table 9 in our revised paper.
>
> **W2: Pruning criteria are overly simplistic.**
>
> We appreciate the reviewer's thoughtful suggestion regarding utilizing PRMs. We would like to clarify our design rationale:
>
> **(1)** The jailbreaking trajectories during training are highly non-monotonic, where seemingly unpromising branches can succeed through late-turn pivots. Greedily selecting seemingly promising trajectories based on intermediate heuristics could **hurt exploration**, which results in **poor diversity** and makes the policy biased toward narrow attack patterns. In our experiments, we observe that if the trajectory diversity is low in the early training stage, the final test performance will drop significantly.
>
> **(2)** Our random subsampling encourages exploration and enables the model to discover diverse attack strategies (demonstrated in Section 5 and Appendix G.1). In addition, from the results in Table 2, our method achieves substantial performance gain (25% ASR on average), which indicates that our pruning is sufficient to enable the attacker to effectively explore diverse and successful attack trajectories.
>
> **(3)** We acknowledge that adopting PRMs is an interesting idea. However, it remains an open and challenging research problem. In this work, we focus on building the first multi-turn RL algorithm for red-teaming, and leave the exploration of PRMs and more sophisticated pruning mechanisms for future work.
>
> **W3: Justification for adaptive masking is unclear.**
>
> We thank the reviewer for raising this question. We would like to clarify that
>
> **(1)** While the reward is not directly tied to format, the penalty applied to negative trajectories—aimed at preventing the generation of such trajectories—also suppresses the production of correct format tokens present in those trajectories. This results in negative gradient updates on the format tokens, causing the model to forget the structured output format learned through the SFT stage. Such negative trajectories are dominant in the early RL training stage, making the unlearning phenomenon more severe.
>
> **(2)** To address this, we disable gradient updates for format token parameters in negative trajectories, while encouraging updates for those in positive trajectories—an approach we refer to as adaptive masking. We have shown in Figure 2 that this adaptive masking strategy, compared with No Masking and Static Masking (disabling the learning of format tokens for both positive and negative trajectories), can significantly reduce the percentage of malformed trajectories and improve the training stability. We have made it clearer in our revision in Lines 194-198.

---

> ### Author Response · Authors · 2025-11-26
> **Official Responses to Reviewer uVB4 [2/2]**
>
> **W4: Reward and evaluation inconsistency.**
>
> During training, we use HarmAugGuard as the reward model since it offers a great trade-off between accuracy and latency (see Appendix D.1 for detailed experiments). During evaluation, we follow the standard protocol in prior works (Chao et al., 2024; Qi et al., 2024; Ren et al., 2024b) and use GPT-4o as a held-out judge for harmfulness assessment (see Lines 254-256). We would like to clarify that there won’t be misalignment issues when adapting different models to the reward model in training and the judge model in the test, and these two models do not need to be the same. Instead, using GPT-4o as the held-out judge avoids the potential reward hacking issue, where the learned policies could exploit specific quirks of the imperfect reward model [1,2]. Besides, it would be computationally prohibitive if we adopt GPT-4o as the reward model during training.
>
> **W5: Ambiguity in tree rollout vs. GRPO comparison.**
>
> We apologize for the confusion in our presentation of Table 2. We would like to clarify that the “standard GRPO” baseline ("w/o tree rollout") is also equipped with our format and topic pruning. The conclusion in Line 370, i.e., “ tree rollout mechanism itself provides 9.8% ASR gain”, is drawn by comparing “w/o tree rollout” in the fourth row and DialTree-RPO in the first row. The result of “w/o any pruning” (in the 3rd row) cannot be directly compared with “w/o tree rollout” (in the 4th row), so it cannot indicate whether GRPO performs better than the tree-rollout-only baseline. We reorganize the results in Table 2 as follows. We have updated the table and the discussions in our revised draft in Lines 371 and 377.
>
> **Table B: Updated Ablation Study on Tree Rollout, as well as Format and Topic Pruning.**
>
> | Method | Tree Rollout | Format Pruning | Topic Pruning | Llama-3.1-8B | Mistral-7B | Gemma-2-9B |
> |--------|:------------:|:--------------:|:-------------:|:------------:|:----------:|:----------:|
> | **DialTree-RPO (Ours)** | ✓ | ✓ | ✓ | **71.5** | **86.5** | **70.5** |
> | - w/o format pruning | ✓ | ✗ | ✓ | 62.5 | 75.5 | 58.5 |
> | - w/o topic pruning | ✓ | ✓ | ✗ | 61.5 | 83.0 | 63.5 |
> | - w/o any pruning | ✓ | ✗ | ✗ | 47.0 | 69.5 | 37.0 |
> | - w/o tree rollout | ✗ | ✓ | ✓ | 59.0 | 81.0 | 59.0 |
>
> **W6: Utterance length limitation.**
>
> We would like to clarify that during our training rollouts, both the attacker’s and target’s output tokens never exceed our set length limits. We selected the 256-token (attacker) and 150-token (target) limits after extensive preliminary experiments to ensure all utterances remain complete. Throughout our entire training process, we logged every generated utterance and verified that no format tokens were truncated due to exceeding the length limit. The malformed outputs shown in Figure 2 stem from the model failing to generate the required format tokens.
>
> **Q1: Tree rollout path.**
>
> No, we do not conduct supplemental sampling when the actual number of trajectories drops after pruning. From Figure 2(c), the percentage of pruned trajectories is mostly less than 40% throughout the whole RL training stage, indicating our approach can maintain a reasonable amount of valid trajectories for effective policy optimization. We have clarified it in our updated draft in Lines 243-244.
>
> **Q2: Jailbreak-R1 implementation.**
>
> Yes, Jailbreak-R1 is a single-turn method as we described in Lines 262-263. We followed its official implementation in our experiments and did not adapt it to multi-turn settings. For fair comparison, we conduct 5 independent jailbreaking attempts per instance using Jailbreak-R1, and mark an instance as successfully attacked if at least one attempt succeeds.
>
> ---
>
> **References**
>
> [1] Defining and Characterizing Reward Hacking. NeurIPS 2022.
>
> [2] Scaling Laws for Reward Model Overoptimization. ICML 2023.

---

> > ### Comment · Reviewer_uVB4 · 2025-11-27
> >
> > Thanks for your detailed clarification. However, I have two follow-up questions: (1) If the outputs never exceed these limits during training, why impose the 256/150 token constraints at all? (2) You mentioned that 256 tokens were selected after experiments showed it worked best, suggesting this limit affects results. But if outputs never actually reach this limit, how does changing this value impact performance?

---

> > > ### Author Response · Authors · 2025-11-27
> > >
> > > Thank you for the thoughtful follow-up question. We would like to address your questions as follows:
> > >
> > > ***Why do we set the max token length limit?*** We impose the token limit as an implementation-level **precaution** against unexpected edge cases to improve the robustness of our training script. When we were exploring algorithm designs or running with suboptimal training configurations for ablation studies, the model may occasionally generate excessively long outputs. While such cases are extremely rare (<0.4% of training instances), their occurrence could immediately cause GPU out-of-memory issues and interrupt training. Such a token limit precaution is particularly crucial as we need to launch many training experiments with limited computing resources.
> > >
> > > ***Why do we set the token limit numbers to 256/150?*** In our preliminary experiments exploring different RL configurations, we observed that the vast majority of rollout trajectories fall within 256 tokens for the attacker and 150 for the target. We selected these limits to strike a balance between accommodating the vast majority of valid outputs and maintaining memory efficiency within our available computation.
> > >
> > > ***Will the max length limit impact the performance of our DialTree-RPO?***  We clarify that the imposed token limit will not impact the performance of our DialTree-RPO. For our finalized method and experiments reported in the paper, we meticulously inspected the training logs and confirmed that all generated outputs of DialTree-RPO fell within the limits. Specifically, the attacker’s averaged output token length is 156.72 (with max=**238** and min=91), and the attacker’s averaged output token length is 106.45 (with max=**141** and min=6). Therefore, the length constraints were never triggered and did not affect the performance of our main method.
> > >
> > > Thank you again for your time and effort in providing valuable feedback that helps strengthen our work. We hope our clarification sufficiently addresses your follow-up questions. Please let us know if you have any further comments.

---

### Author Response · Authors · 2025-12-03
**General Responses to the AC**

We thank the reviewers for taking their time to provide constructive comments to improve our paper. We are pleased to note that all the reviewers recognized the value of our work, as evidenced by the following positive comments:
* "The paper addresses an important gap in AI safety" / "The problem of RL post-training in a multi-turn setting is sound and important" (**uVB4**; **63Yg**)
* "It is also interesting to regard the multi-turn setting as a form of decision-making in dialogue in which RL can be applied" / "extending GRPO to dialogues with non-verifiable rewards" (**hWje**; **HvSg**; **63Yg**)
* "The tree-based dialogue exploration and pruning mechanisms are conceptually well-motivated… which is crucial for efficient red-teaming" / "enables structured exploration of exponentially large attack spaces" (**uVB4**; **hWje**)
* "adaptive masking solves the critical format unlearning problem" (**hWje**)
* "introduces a framework that eliminates the need for human-written templates" (**HvSg**)
* "The empirical results are strong and comprehensive" / "Tests against diverse closed-source and open-source models; includes thorough ablations" / "validates automated evaluation with human agreement"  (**uVB4**; **hWje**)
* "26% better than the prior best—with only 3 queries per attack; experiments are thorough and reproducible" (**HvSg**)
* "The proposed method has promising transferability" (**uVB4**)
* "Exposes a systemic LLM vulnerability to strategic multi-turn attacks; releasing the methodology accelerates the development of multi-turn safety defenses" (**HvSg**)
* "The paper is generally well-organized, and the figures are clear to follow" (**63Yg**)

The reviewers' thoughtful feedback has substantially improved our paper. During the rebuttal, we carefully addressed all reviewers' comments in our responses, which were recognized by the reviewers. Particularly, **Reviewer 63Yg explicitly raised their score from 4 to 6**, stating "My concerns have been addressed" before the author-reviewer discussion ended due to the incident. Reviewer hWje maintained a positive assessment at 6. We have also thoroughly addressed Reviewer uVB4's comments through follow-up discussion. Based on the reviewers' suggestions, we have updated the paper with the following major revisions and **marked them in blue**:
* **In response to uVB4 and HvSg's request for the SFT baseline**, we compared the SFT-only baseline with our full DialTree-RPO across 10 target models and observed clear, substantial gains from the RL stage. We further demonstrated generalizability by applying our framework to alternative RL algorithms (e.g., DAPO), which achieved similarly strong improvements.
* **In response to hWje's request for comparisons with additional state-of-the-art jailbreaking methods and evaluation on Claude Sonnet**, we added experiments with ActorAttack and X-Teaming and found that DialTree-RPO significantly outperforms both methods. We also evaluated Claude-4-Sonnet, demonstrating that our method retains strong effectiveness even against one of the most robustly safety-aligned LLMs.
* **In response to uVB4 and 63Yg's questions about adaptive masking**, we revised Eq. (3) and (4) to formally define adaptive masking and clarified how it prevents negative gradient updates on format tokens in negative-advantage trajectories and addresses the format unlearning issue in multi-turn RL.
* **In response to 63Yg's questions about reproducibility and exploration diversity**, we ran five independent training runs with different random seeds for all main experiments. The results confirm that our findings are consistent and reproducible across multiple runs. We also added trajectory diversity analysis using Self-BLEU scores, showing that our approach can substantially improve trajectory diversity.
* **In response to HvSg's question about robustness under defenses**, we conducted additional experiments with systems where target models are equipped with external guardrails. Our method maintains high ASR even with guardrails, demonstrating robustness against additional safety mechanisms.
* **In response to uVB4's questions about Table 2 presentation and token limits**, we reorganized the ablation table to clarify comparisons between tree rollout and standard GRPO. We clarified that the 256/150 token limits serve as implementation-level precautions and verified through training logs that all outputs in our main experiments fell within these limits without truncation.
* **In response to ethics concerns raised by hWje and HvSg**, we detailed our responsible disclosure practices, i.e., the trained models, code, and data will be provided only to verified researchers with legitimate safety research objectives who agree to use resources exclusively for defensive applications without redistribution.

We hope our responses provide AC with a clear overview of how we have thoroughly addressed all reviewers' concerns and improved the paper.

Best regards,

The Authors

---

### Meta-Review · Area_Chair_mxfr · 2026-01-07

**Summary:**

This paper proposes DialTree-RPO, an on-policy reinforcement learning framework integrated with tree search, which autonomously discovers diverse multi-turn adversarial attack strategies by treating dialogue as a sequential decision-making problem.

The reviewers' primary technical concerns focused on verifying the independent gains of the RL stage, providing additional baseline comparisons, clarifying the specific roles of the algorithm's internal components, and demonstrating robustness under defense mechanisms. The authors successfully addressed these concerns by adding supplementary experiments. In addition, there were two subjective concerns regarding a perceived lack of novelty and insufficient data contribution.

Among the four reviewers, two explicitly provided positive evaluations. Reviewer uVB4, who assigned the lowest score, demonstrated a satisfied attitude during the rebuttal and is highly likely to raise their score. Regarding Reviewer HvSg, who explicitly stated they would not raise their score, while I believe the authors have effectively resolved the technical aspects of their concerns.

Therefore, AC is to recommend acceptance.

**Reviewer Concerns:**

The authors have effectively addressed all reviewer concerns by providing new experimental evidence that demonstrates the significant advantage of the proposed algorithm over the SFT-only baseline and verifying its robustness under various defense mechanisms, including external guardrails. Furthermore, the authors validated the critical contributions of Tree Rollout and Pruning through comprehensive ablation studies, demonstrated superior performance against additional baselines such as ActorAttack and X-Teaming.

**Reviewer Scores:**

Reviewer uvb4: I anticipate that this reviewer would raise their initial score of 2. The authors successfully addressed the six weaknesses and two questions raised in the first round of reviews through their rebuttal. Furthermore, the reviewer raised a minor engineering follow-up question in the second round, which was also satisfactorily answered by the authors. Therefore, I believe it is highly likely that this reviewer would increase their score.
Reviewer 63Yg: This reviewer has explicitly stated their intention to raise their score to 6.
Reviewer hWje: The authors included the comparison methods requested by this reviewer and achieved excellent experimental results. I believe this reviewer would at least maintain their original score of 6, with a reasonable possibility of increasing it.
Reviewer HvSg: Although the authors provided the supplementary experiments, this reviewer explicitly stated that they plan to maintain their initial score of 4.

---

### Decision · Program_Chairs · 2026-01-26

Accept (Poster)